# The Perfect Combination: Enhancing Patient Response to PD-1-Based Therapies in Epithelial Ovarian Cancer

**DOI:** 10.3390/cancers12082150

**Published:** 2020-08-03

**Authors:** Nicole E. James, Morgan Woodman, Paul A. DiSilvestro, Jennifer R. Ribeiro

**Affiliations:** 1Program in Women’s Oncology, Department of Obstetrics and Gynecology, Women and Infants Hospital, Providence, RI 02905, USA; nromano@wihri.org (N.E.J.); mwoodman@wihri.org (M.W.); pdisilvestro@wihri.org (P.A.D.); 2Department of Obstetrics and Gynecology, Warren Alpert School of Medicine, Brown University, Providence, RI 02903, USA

**Keywords:** PD-1, ovarian cancer, immunotherapy

## Abstract

Epithelial ovarian cancer (EOC) is the most lethal gynecologic malignancy, with an overall 5-year survival of only 47%. As the development of novel targeted therapies is drastically necessary in order to improve patient survival, current EOC clinical trials have heavily focused on immunotherapeutic approaches, centered upon programmed cell death 1 (PD-1) inhibitors. While PD-1 monotherapies have only exhibited modest responses for patients, it has been theorized that in order to enhance EOC patient response to immunotherapy, combinatorial regimens must be investigated. In this review, unique challenges to EOC PD-1 response will be discussed, along with a comprehensive description of both preclinical and clinical studies evaluating PD-1-based combinatorial therapies. Promising aspects of PD-1-based combinatorial approaches are highlighted, while also discussing specific preclinical and clinical areas of research that need to be addressed, in order to optimize EOC patient immunotherapy response.

## 1. Introduction

Epithelial ovarian cancer (EOC) is a highly fatal gynecologic malignancy, with an estimated 13,940 predicted deaths in the United States in 2020 [1]. This high lethality is attributed to the fact that the majority of patients are diagnosed at an advanced stage, and eventually develop chemoresistant disease following front-line treatment [2]. While approximately 80% of patients achieve remission following initial therapy, a majority will relapse within 16–18 months, highlighting the urgent need for novel targeted therapies [3]. In recent years, advances such as the development of anti-angiogenic and poly ADP-ribose polymerase I (PARP) inhibitors have modestly improved patient progression-free survival (PFS) [4,5,6,7], but a demand still exists for innovative EOC therapeutics to improve long-term outcomes. As across all cancers, EOC clinical trials currently focus on immunotherapy, with a strong emphasis placed on the most well-known immune-target, programmed cell death 1 (PD-1). 

PD-1 is expressed on CD8+ T cells, B cells, and natural killer (NK) cells during chronic antigen exposure. Upon PD-1 binding to its ligands, programmed cell death ligand 1 or 2 (PD-L1/PD-L2) on tumor cells, T cell receptor (TCR) signaling and CD28 co-stimulation is inhibited, ultimately leading to effector cell inactivation [8,9]. Thus, upregulation of PD-L1 on cancer cells provides a protective mechanism to the tumor by shielding it from T cell infiltration [10]. Recent research has also shown that PD-L1 expressed on dendritic cells further contributes to T cell inactivation [11], highlighting the necessity for further research into the complexities of the PD-1/PD-L1 axis. 

Within the past decade, there has been an explosion of clinical research on immunotherapies targeted against PD-1 (pembrolizumab, nivolumab, cemiplimab) and PD-L1 (atezolizumab, durvalumab, avelumab), which have revolutionized cancer care. The first inhibitors, pembrolizumab and nivolumab, were approved by the FDA for unresectable stage III/IV metastatic melanoma in 2014 [12,13]. In addition to metastatic melanoma, PD-1 therapy has also been FDA indicated for non-small cell lung cancer [14], head and neck squamous cell carcinoma [15], renal cell carcinoma, and urothelial cancer [16]. Unfortunately, in EOC, PD-1-based monotherapy has not been nearly as successful. EOC phase I trials of PD-1 inhibitors demonstrated an objective response rate (ORR) between 10–15%, with disease controlled in fewer than half of all patients [17,18,19,20]. These results indicate that PD-1 therapy alone will not be efficacious enough to improve survival for EOC patients. However, evidence is emerging that combinatorial methods will be critical to optimizing EOC patient response to immunotherapy [21].

This review will discuss specific challenges that impede EOC patient response to PD-1-based therapies, and how these barriers can be potentially mitigated through combinatorial methods. Results from emerging preclinical and clinical studies combining PD-1/PD-L1 inhibitors with current standard of care and investigational EOC therapies will be presented, while also highlighting the deficiency in key mechanistic and biomarker studies that are necessary in order to further enhance efficacy of these combinatorial regimens. 

## 2. Challenges to PD-1 Therapy 

As ovarian cancer is considered an immunoresponsive cancer, there have been multiple hypotheses as to why PD-1-based therapies have not been as efficacious as in other cancers. It has been theorized that poor EOC patient responses are due to the unique tumor microenvironment (TME), higher levels of regulatory T cells (T_regs_) in ovarian tumors, and barriers in identification of patients who will benefit most from PD-1 therapy. 

### 2.1. Unique Tumor Microenvironment

The unique components of the ovarian peritoneal TME greatly challenges immunotherapy response [22]. Malignant ascites produced by the peritoneal cavity are rich in cytokines and growth factors that not only promote tumor growth, but also impair the ability of NK lymphocytes to function properly as effector cells [23]. Furthermore, these impaired NK cells express PD-1 and exhibit a significantly reduced ability to kill PD-1+ tumor cells and secrete IFNγ and TNFα [24]. In addition to the tumor promoting properties of the ascites, the omentum represents a preferential site for ovarian cancer metastasis that occurs via both peritoneal and vasculature spread, likely due to a variety of factors including cytokine signaling, receptor binding, and the highly vascularized adipose tissue itself [25]. During omental metastasis, ovarian cancer cells are trapped in “milky spots”, which are vascularized regions containing immune cells with both innate and adaptive immune capabilities [25]. Adipocytes are present at high levels within the omentum and secrete IL-6, IL-8, monocyte chemoattractant protein 1 (MCP-1), tissue inhibitor of metallopreoteinases-1 (TIMP-1), and adiponectin to promote tumor growth and metastasis. Furthermore, adipocytes provide a rapid source of fuel for the tumor by altering lipid metabolism [26]. In an ovarian cancer mouse model, obese mice developed increased metastatic tumor burden, which was attributed to increased vascularity, altered regulation of the fatty acid pathway, and a decreased M1/M2 macrophage ratio [27].

In addition to a high level of adipocytes, cancer-associated fibroblasts (CAFs) in EOC are also present in the ovarian TME and promote immunosuppression through the secretion of numerous growth factors and cytokines. Specifically, CAFs produce elevated levels of hepatocyte growth factor (HGF), which increases tumor cell proliferation, invasion, migration, and chemoresistance [28]. In addition, CAFs promote increased levels of other cytokines such as COX-2, CXCL1, CCL5, CXCL11, and IL-6, which in turn promote proliferation and the epithelial-mesenchymal transition [22]. A study by Givel et al. identified one subset of CAFs in EOC, CAF-S1, which is responsible for the recruitment of T_regs_ to the tumor site. Moreover, these CAF-S1 cells secrete CD73, B7-H3, and IL-6, which encourages survival and proliferation of these intratumoral T_regs_ [29]. In addition to their influence on epithelial and immune cells, CAFs can directly stimulate angiogenesis by acting on endothelial cells via secretion of vascular endothelial growth factor (VEGF)-C [30]. 

Furthermore, CAFs are the primary source of extracellular matrix components, such as collagen. Fibrosis is likely an important mediator of ovarian tumorigenesis, which is supported by recent research showing that use of metformin, which treats and prevents fibrosis, is associated with an 88% reduction in ovarian cancer incidence in women taking it for the treatment of type II diabetes [31]. In addition, the presence of age-associated fibrosis, which had been previously demonstrated in mice [32], was recently confirmed by McCloskey and colleagues in a cohort of human samples [31]. This age-associated fibrosis may explain the pro-tumor niche of the ovary that supports the growth and spread of tumor cells originating from the fallopian tubes [31].

The highly fibrotic microenvironment of ovarian cancer also likely contributes to poor immunotherapeutic responses. The physical resistance provided by the basement membrane to immune cells is high given its intermolecular interactions and structure, which is significantly more stiff and rigid than the loose networks of collagen fibrils making up the interstitial matrix. Importantly, aberrant modeling of the ECM leads to increased tissue stiffness, and solid tumors display increased rigidity compared to normal tissue counterparts [33]. Tissue modulus (stiffness) values were higher in high disease score biopsies of high grade serous EOC, and there were significant correlations between malignant cell area, stromal area, and disease score of EOC biopsies [34]. The increased stiffness of tumor tissue is of vital importance when considering how T cells infiltrate tumors. While there is data lacking in ovarian cancer, it has been shown in lung cancer that T cell infiltration occurs in loose fibronectin/collagen regions but is impaired in dense matrix fibers, leading to preferential stromal T cell accumulation and restricted tumor infiltration [35]. In another study, lack of response to PD-L1 inhibition occurred particularly in patients with tumors showing preferential localization of CD8+ T cells in the collagen-rich stroma as opposed to the parenchyma in urothelial cancer [36]. Finally, another study found that high levels of stromal serum biomarkers predicted poor response to CTLA4 checkpoint blockade in melanoma [37]. These results are important to view in the context of what is known about CD8+ T cell infiltration in ovarian cancer. Studies have shown that specifically intra-epithelial associated, not stromal associated, CD8+ T cells are related to improved outcomes for ovarian cancer patients [38,39,40], suggesting that CD8+ T cells “sequestered” in the stroma are restricted from accessing and killing the tumor cells. Further research is needed to understand how the fibrotic environment of ovarian tumors may specifically contribute to ovarian tumor resistance to immunotherapy. 

In summary, the complex interplay between the stromal, epithelial, immune, and endothelial cells within the ovarian TME highlight the need for combinatorial approaches to overcome a highly immunosuppressive milieu.

### 2.2. Elevated Levels of Highly Activated T Regulatory Cells

The elevated number of highly active T_regs_ in ovarian tumors represents another significant barrier to the effectiveness of anti-PD-1 therapy. T_regs_ in tumors or ascites of EOC patients are more proliferative and activated compared to matched blood derived T_regs_ [41,42]. Ascitic T_regs_ were shown to be skewed toward an effector/memory state [42], which displayed frequent expression of the immunoglobin superfamily member CD147 [42], a marker of activated Tregs [43] that also promotes ovarian tumorigenesis through activation of VEGF and MMP-9 secretion [44]. Furthermore, Bu and colleagues found that suppressive function of tumor infiltrating T_regs_ was dependent upon expression of the co-inhibitory receptor T cell immunoglobulin and mucin-domain containing-3 (TIM-3) [41].

It has been proposed that the phenotype of T_regs_ in EOC could be key to understanding why the response rate to PD-1 therapy has been low. In a study by Toker et al., ovarian cancer tumor infiltrating T_regs_ exhibited a highly activated phenotype distinct from T_regs_ in melanoma, a cancer with exceptionally high response rates to PD-1 therapy. Using mass cytometry and transcriptomic analyses, Toker and colleagues discovered that ovarian cancer T_regs_ exhibited increases in PD-1 and 4-1BB levels, which are associated with greater suppressive capabilities. In comparison, melanoma T_regs_ exhibited lower levels of FOXP3, PD-1, and 4-1BB and were less potent suppressors of cytotoxic T cell proliferation [45]. Taken together, these studies highlight how the high number and activated phenotype of EOC T_regs_ may create an environment particularly resistant to immunotherapy.

### 2.3. Patient Selection Barriers

PD-L1 expression on tumor cells and tumor associated immune cells has been used as a predictive biomarker for EOC patient response to PD-1 blockade [46]. However, PD-L1 as a biomarker for patient selection in EOC is problematic, due to the fact that it fails to capture all patients who will respond to PD-1 therapy [47]. In addition, tumoral expression of PD-1 and PD-L1 and its relation to EOC patient immunotherapy response remains controversial [48]. Early EOC Phase I trials of PD-1 monotherapies found no correlation between clinical response and PD-L1 expression in tumor tissue [17,49]. A potential explanation for this phenomenon is that PD-L1 exhibits low expression in EOC, with just 10–33% of patient tumors characterized as PD-L1 positive. This finding suggests that not all EOC tumors use the PD-1/PD-L1 pathway to achieve immune evasion [48], highlighting that this pathway should be targeted in combination with other agents to enhance patient response to immunotherapy [50].

Furthermore, the prognostic significance of PD-L1 in EOC is debatable. Several studies have reported that high levels of PD-1 or PD-L1 are indicators of favorable prognosis [51,52,53,54]. Specifically, Webb et al. found PD-L1 to be associated with higher levels of tumor infiltrating lymphocytes (TILs). The group reported that PD-L1+ cells and CD8+ TILs together had a greater association with improved survival compared to CD8+ TILs alone, and PD-L1 was associated with both cytolytic and suppressive gene products at the transcriptional level [52]. Conversely, Chatterjee et al. found that low levels of PD-1 on lymphocytes is associated with improved survival [55]. Moreover, a study by Hamanishi et al. determined that patients with a high expression of PD-L1 exhibited a worse prognosis then patients with a lower expression, and observed an inverse correlation between PD-L1 and intraepithelial CD8+ TILs [56]. Interestingly, Drakes et al. found that while both PD-1 and PD-L1 tumoral expression are significantly correlated with advanced disease, no association was detected between PD-1/PD-L1 expression and survival outcomes [57]. Finally, a large meta-analysis of 1630 ovarian tumors found no correlation between PD-L1 and PFS or OS. However, subgroup analysis showed that in EOC patients from Asian countries, PD-L1 expression is significantly associated with poor OS and PFS, while in non-Asian countries, PD-L1 expression was associated with favorable outcomes [58]. Taken as a whole, these inconsistencies highlight the challenges of using PD-L1 as a predictive or prognostic indicator for EOC, and further emphasizes the inadequacies of PD-1-based monotherapies. 

## 3. PD-1 Based Combinatorial Approaches 

### 3.1. PD-1 and Chemotherapy

#### 3.1.1. Preclinical Studies

Many studies in EOC have reported an upregulation of PD-1/PD-L1 following chemotherapy [59,60,61,62], providing a strong justification for this combinatorial treatment. Peng et al. uncovered that the induction of PD-L1 post-chemotherapy was mediated by nuclear factor kappa-light-chain-enhancer of activated B cells (NF-κB) [62]. Furthermore, they observed an upregulation of both PD-L1 and major histocompatibility complex I (MHC I) resulting from treatment with either gemcitabine or paclitaxel, which was dependent upon the upregulation of NF-κB. The authors excluded the influence of IFNγ in the upregulation of PD-L1 and MHC I, as both were still increased following chemotherapy treatment in IFNγ knockdown cell lines [62]. Lastly, in an in vivo ID8 syngeneic model, mice treated with either a PD-1 or PD-L1 blockade and paclitaxel exhibited a longer survival than mice treated with chemotherapy alone [62]. In agreement with these results, another study determined that nivolumab combined with cisplatin reversed cisplatin resistance in A2780/DDP ovarian cancer cells [63]. The group found that the combinatorial treatment significantly increased cancer cell apoptosis and significantly decreased expression levels of ADAM17 [63], a metallopeptidase also known as TNFα converting enzyme [64]. However, limited conclusions can be drawn from this data as PD-1 was targeted in a cell culture system devoid of an immune component. Finally, a study by Xu et al. determined that microRNA-424(322) expression reverses chemoresistance in EOC through the inhibition of PD-L1, leading to an increase in CD8+ T cells, and a decrease in myeloid derived suppressor cells (MDSCs) and T_regs_ [65]. The authors hypothesize that regulation of PD-L1 via microRNA-424(322) is potentially responsible for the mechanistic relationship between PD-L1 and chemoresistance [65]. These studies highlight the diverse mechanisms by which PD-1 therapy may exert its therapeutic effects in ovarian cancer.

In addition to traditional platinum-based frontline therapies, PD-1 has also been tested in combination with trabectedin, an alkylating chemotherapeutic agent [66], in EOC. Combined with an anti-PD-1 antibody, trabectedin was found to exert a strong antitumor response, with half of all mice achieving a complete response [67]. In addition, the dual treatment increased CD4+ and CD8+ T cells while also decreasing T_regs and_ MDSCs. Furthermore, the group observed an increase in Th1 effector T cell recruitment, demonstrated via increased transcript levels of IFNγ, IL-12p40, and T-bet [67]. Similarly, a study by Germano et al. reported a decrease in immunosuppressive factors such as MDSCs and tumor- associated macrophages (TAMs) from in vivo combinatorial treatment with trabectedin and anti-PD-1 therapy [68]. These results are encouraging, as trabectedin is known to have immunomodulary effects, such as inhibition of IL-6 and TAMs [66,69], and therefore should synergize well with immune checkpoint blockades in EOC [70].

Further immunotherapeutic agents have been found to enhance the combinatorial treatment of chemotherapy and a PD-1/PD-L1 blockade. Ghaffari and colleagues found that addition of a stimulator of interferon genes (STING) agonist to an anti-PD-1 antibody and carboplatin led to the longest survival in an ovarian ID8 p53-null in vivo mouse model, with the highest level of intratumoral PD-1+ cells found in mice treated with the STING agonist [71]. Furthermore, Hartl et al. determined that therapy with STING agonist 2’3’cGAMP, anti-IL-10, and PD-L1 therapy with frontline chemotherapy increased survival in an ID8 VEGF Defb29 mouse model [72]. This therapeutic regimen also significantly reduced MDSCs while increasing infiltrating T cells and dendritic cells (DCs) [72]. Interestingly, the authors noted that dosing and timing matters, with immunotherapies needing to be given simultaneously with paclitaxel and carboplatin to reverse chemotherapy induced immunosuppression. Mechanistically, the authors determined that highly activated CD4+ effector T cells that exhibited granzyme B mediated toxicity along with elevated levels of ICOS, CD44, and PD-1 were responsible for the anti-tumoral effects of the therapeutic regimen [72]. Additionally, a study by Wahba et al. reported that treatment with a PD-1 inhibitor in combination with paclitaxel and chimeric antigen receptor (CAR) T cells targeting ErbB homo-and hetero-dimer (T4 immunotherapy) promoted a synergistic reduction in tumor burden, compared to either treatment alone [60]. To address the safety profiles in humans when combining multiple immunotherapies and chemotherapy, further clinical investigations are warranted. 

Conversely, not all chemotherapy and PD-1-based treatment combinations have been found to be efficacious in EOC. Grabosch et al. found that in vivo targeting of PD-L1 in combination with cisplatin did not substantially impact survival, despite having the ability to control tumor burden. The authors postulated that dosing and timing changes of anti-PD-L1 therapy may improve response [59]. In addition, Zhu et al. found that mice treated with carboplatin alone or with dual treatment of carboplatin and anti-PD-L1 therapy both displayed increased peritoneal CD4+ and CD8+ T cells and decreased T_regs_ and MDSCs. Furthermore, mice treated with carboplatin alone exhibited the greatest survival [50]. These studies highlight the inconsistencies in the data involving the efficacy of combinatorial approaches of chemotherapy and PD-1-based therapies in EOC, and call for comprehensive studies that examine multiple chemotherapeutic agents and dosing of anti-PD-1/PD-L1 agents. 

#### 3.1.2. Clinical Studies 

There has been evidence in EOC to suggest that chemotherapy treatment can enhance immune activation in patient tumors. Böhm et al. profiled pre- and post-neoadjuvant chemotherapy (NACT) omental biopsies and found significantly higher levels of PD-1, PD-L1, CTLA-4, and T cell activation markers in post-treatment biopsies [61]. This data strongly suggests that EOC patients would benefit from the addition of anti-PD-1 therapy in the neoadjuvant setting. However, more studies like this are crucial in order to identify which patients will most benefit from the addition of immunotherapy in the frontline setting. In EOC, there have been a handful of trials seeking to determine the efficacy of adding a PD-1 or PD-L1 inhibitor to frontline and recurrent chemotherapy (Table 1). A large phase III study, JAVELIN OVARIAN 100, was initiated in 2016 to evaluate if avelumab added to frontline paclitaxel and carboplatin, followed by maintenance avelumab could improve PFS (NCT02718417). However, due to lack of efficacy concluded from interim analysis, this study was terminated. Nonetheless, there are still two open phase II trials in EOC examining pembrolizumab combined with frontline chemotherapy, followed by maintenance pembrolizumab (NCT02766582, NCT02520154). Although PFS results for these two frontline chemotherapy trials are currently pending, it is revealing that the JAVELIN OVARIAN 100 trial was discontinued, ultimately suggesting that adding a PD-1/PD-L1 inhibitor to frontline chemotherapy will not be efficacious. Besides comparing PFS resulting from frontline treatment, there is one ongoing trial evaluating pathologic objective response rate (pORR) after NACT and pembrolizumab treatment (NCT02834957). Therefore, it is still undetermined if the addition of a PD-1/PD-L1 inhibitor to NACT can affect pathologic response in EOC. Furthermore, an ongoing matched pharmacodynamics trial is investigating durvalumab combined with frontline chemotherapy with the goal of determining effects of this regimen on molecular markers within immune-related pathways (NCT02726997). Results from this trial could potentially yield key mechanistic information that can be exploited to improve EOC patient response to frontline therapy. 

In the recurrent setting, PD-1-based therapies have been studied in combination with gemcitabine and cisplatin (NCT02608684), pegylated liposomal doxorubicin (PLD) (NCT02580058, NCT02865811), carboplatin (NCT03029598), dose-dense paclitaxel (NCT02440425), and carboplatin with cyclophosphamide (NCT02914470). While many of these studies are ongoing, there have been a handful of interval results reported. One phase II study evaluating the combination of pembrolizumab with gemcitabine and cisplatin observed that out of 18 total patients, one patient achieved a complete response (CR) (5.6%), ten patients exhibited a partial response (PR) (55.6%), five patients (27.8%) had stable disease (SD), and two patients (11.1%) progressed (NCT02608684), highlighting promise for this combinatorial regimen. A large phase II trial, JAVELIN OVARIAN 200, examined the efficacy of avelumab in combination with PLD (NCT02580058). This trial did not meet its primary endpoint, as the dual regimen did not significantly improve PFS or OS compared to PLD alone, despite demonstrating clinical activity. Furthermore, analyses suggested that there was a greater improvement in PFS and OS in the combinatorial group versus PLD alone in patients with PD-L1+ tumors [73], further convoluting the use of PD-1 as a marker of patient immune response. In addition, a phase II trial examining the combination of pembrolizumab and PLD in the recurrent setting (NCT02865811) reported an objective response rate (ORR) of only 11.5% (three out of 23 total patients) and a higher rate of pneumonitis, compared to other recurrent therapeutic regimens. However, results from the study’s primary endpoint, clinical benefit rate, are still pending [74]. Interestingly, Inayama et al. reported two cases of palliative care patients that had an unexpected response with nivolumab and chemotherapeutic agents, reported from a follow-up phase II study of nivolumab (UMIN00005714) [75]. In one patient experienced PR following treatment with PLD and nedaplatin, a derivative of cisplatin [76]. In a second patient, PLD also produced a PR, with nedaplatin resulting in SD. Both patients were alive post-follow up, inferring that nivolumab has the potential to sensitize patients to chemotherapeutic agents [75]. These results highlight the critical need to improve upon identification methods that select exceptional PD-1-based therapy responders. 

Overall, results from combinatorial trials involving PD-1/PD-L1 inhibitors and chemotherapy suggest that this regimen in the frontline setting will not be efficacious, with more promise in recurrent lines. In addition, in order to improve the efficacy of this combination, many trials have been initiated that also include angiogenic and PARP inhibitors, which will be discussed in the following sections. Moreover, as more preclinical and clinical studies are performed to better elucidate synergistic mechanisms between chemotherapy and PD-1/PD-L1 inhibitors, it is hoped that that this understanding will improve therapeutic efficacy of PD-1-based therapy and chemotherapy in EOC.

### 3.2. PD-1 and Anti-Angiogenic Therapy

#### 3.2.1. Preclinical Studies

Interestingly, despite the clinical attention anti-angiogenic and PD-1-based combinations have been given in EOC, there is a crucial gap in the number of mechanistic pre-clinical studies that have been performed. One study by Zeng et al. tested the efficacy of a chemokine receptor 4 (CXCR4) antagonist (AMD3100) in combination with an anti-PD-1 monoclonal antibody in vivo [77] CXCR4’s interaction with its ligand chemokine stromal cell-derived factor-1 (CXCL12) plays a crucial role in promoting tumor invasion and metastasis via promotion of myeloid bone marrow-derived cells, CAFs, and angiogenesis [78]. Dual treatment blocking this axis along with PD-1 significantly reduced tumor burden and increased survival in an ID8 mouse model. Moreover, dual treatment contributed to an immunostimulatory TME, increasing levels of IFNγ, IL-2, and CD4+ and CD8+ effector T cells to a significant degree compared to either monotherapy alone. Lastly, the dual treatment produced significantly more memory T cells and M1 polarization levels, and reduced MDSCs, but had no effect on T_reg_ levels [77]. 

In EOC, there has only been one study examining the efficacy of the most commonly used anti-angiogenic agent, bevacizumab, a monoclonal antibody against VEGF-A with anti-PD-1 therapy. Zhang et al. observed significantly higher levels of VEGF, semaphorin4D (SEMA4D), and PD-L1 in treatment-naïve patient tumors that were later found to be responsive to bevacizumab therapy, versus those that were declared non-responsive. In addition, the authors discovered that atezolizumab and bevacizumab in combination substantially inhibited cisplatin-resistant ovarian cancer cell proliferation, migration and invasion, which they attributed to the impediment of EMT through the signal transducer and activator of transcription 3 (STAT3) pathway. Finally, an in vivo mouse model demonstrated synergistic reduction in tumor burden in the mice treated with atezolizumab and bevacizumab compared to either monotherapy alone [21]. While these results appear promising and suggest non-immune mediated effects of PD-1-based treatments, as this study was not performed in an immunocompetent model, a deficiency remains in our understanding of how these therapies act in concert to promote anti-tumor immunity. In contrast, immunocompetent in vivo studies with combinatorial therapies have been performed in melanoma, pancreatic neuroendocrine tumors, metastatic breast, and small cell lung cancer [79,80], highlighting that ovarian cancer research is lacking in uncovering the mechanistic interplay between anti-angiogenic and PD-1-based therapy. 

#### 3.2.2. Clinical Studies 

There is currently one phase II trial (NCT02873962) with interim results evaluating the efficacy of nivolumab in combination with bevacizumab in recurrent EOC. A total of 38 women were enrolled in this trial, with 18 patients classified as platinum resistant and 20 platinum sensitive [81]. The ORR was found to be 28.9%; however, when broken down by platinum sensitivity, platinum resistant patients exhibited an ORR of 16.7%, and platinum sensitive patients had an ORR of 40%. Remarkably, out of the total of 12 patient responders, ten patients had tumoral PD-L1 less than 1% [81], highlighting the challenge of using PD-L1 as a marker for PD-1 immune response in EOC. While the results reported were encouraging for platinum-sensitive patients, they suggest that different combinatorial approaches, or the addition of a third therapeutic agent, are necessary in order to improve platinum resistant patient response. 

The addition of bevacizumab to the combinatorial regimen of PD-1/PD-L1 inhibitors and chemotherapy is currently being studied clinically. In the frontline setting, a phase III study GOG3015/ENGOT OV39 is investigating the efficacy of combining platinum-based chemotherapy with bevacizumab and atezolizumab, in patients stratified by tumoral PD-L1 positivity (NCT03038100). In the recurrent setting, there are also large phase II/III trials examining the combination of bevacizumab, and atezolizumab with chemotherapy such as PLD, paclitaxel, or carboplatin (NCT03353831, NCT02839707, NCT02891824). Results from these trials are highly anticipated, as they will determine if the combination of PD-1/PD-L1 inhibitors, angiogenic inhibitors, and chemotherapy can improve EOC patient survival. Moreover, these results will allow for further evaluation of PD-L1 as a marker of therapeutic response.

Besides chemotherapy-based trials, there have been additional innovative studies evaluating angiogenic and PD-1/PD-L1 inhibitors. A phase II trial, EORTC-1508 is currently examining the efficacy of combining atezolizumab, bevacizumab, and acetylsalicylic acid in patients who have recurred (NCT02659384), as it has been found preclinically in EOC that acetylsalicylic acid exerts anti-tumoral effects together with VEGF A inhibition [82]. In addition, a phase I trial is seeking to establish the safety of pembrolizumab and nintedanib, a novel angiokinase inhibitor against VEGF receptor (VEGFR 1,3) fibroblast growth factor receptor (FGFR), platelet-derived growth factor alpha and beta receptor (PDGFRαβ), and RET, a proto-oncogene [83] (NCT02856425). Furthermore, there are numerous trials currently investigating the regimen of anti-angiogenic, PD-1/PD-L1, and PARP therapy, which will be reviewed in the next section. 

Largely, many angiogenic and PD-1-based therapeutic combinatorial trials are ongoing. A full list of current trials can be seen in Table 2. As the field awaits results from these clinical investigations, it is also imperative that pre-clinical mechanistic studies be performed to better elucidate if and how these two therapies synergize to affect the EOC TME.

### 3.3. PD-1 and PARP Therapy

#### 3.3.1. Preclinical Studies 

It has been established that EOC patients with breast cancer type 1/2 susceptibility protein (BRCA1/2) mutations possess higher tumoral expression of PD-1 and PD-L1 [84,85], suggesting that these patients, who are already more highly responsive to PARP inhibitors, will also be more receptive to PD-1-based therapy. Since EOC patients with BRCA mutations are most likely to benefit from PARP therapy [86] and also have higher neoantigen loads [84], it is logical that this subset of patients may benefit most from dual PARP and PD-1 blockade. There have been two in vivo studies in EOC that have demonstrated efficacious combinatorial therapy involving PARP and PD-1 inhibition in BRCA-deficient mouse models [87,88]. Ding et al. reported that PD-1 blockade combined with olaparib led to a synergistic reduction in mouse tumor burden [88]. However, the authors noted that despite this marked reduction in tumor growth, mice treated with olaparib alone or in combination with anti-PD-1 therapy ultimately did not improve OS. The authors noted that treatment with olaparib led to higher intratumoral levels of CD4+ and CD8+ T cells, with significantly higher secretions of IFNγ and TNFα. The addition of a PD-1 blockade further enhanced secreted levels of these cytokines on cytotoxic T cells. Interestingly, no significant reduction in T_regs_ was found by single agent or combinatorial treatment. Mechanistically, the group determined that the anti-tumoral immunity by PARP inhibition is propagated via activation of the STING pathway. The authors observed that treatment with olaparib led to increases in macrophages, DCs, and markers of STING activation such as CXCL9, CXCL10, and IFN-β. To demonstrate that this phenomenon was specific to ovarian BRCA-deficient tumors, in vivo experiments were repeated in a BRCA-proficient model, in which no increase in STING activation markers was observed [88]. This finding can potentially be explained by the significant increase of PD-L1 following single agent treatment with olaparib in the BRCA-deficient model, as the unique molecular properties BRCA-deficient tumor possess to promote STING pathway responses still remains unknown [88].

An additional study by Wang et al. observed that mice treated with pembrolizumab and niraparib exhibited a significant reduction in tumor growth, compared to either therapy alone. Furthermore, the authors assessed the robustness of this therapeutic response by including an observation period post dosing, at which point the tumor regrowth rate for the mice treated in combination exhibited no signs of growth, as opposed to the monotherapy treated mice [87]. Additionally, at the end of the observation period, mice that were tumor free from single or combinatorial treatment were re-challenged by tumor cell inoculation. Remarkably, seven weeks after re-inoculation, all mice previously treated in both the single and combinatorial arms had no signs of tumor growth, demonstrating potential immune memory resulting from treatment with either niraparib or pembrolizumab. The authors also found marked tumor reduction following dual treatment in a BRCA-proficient in vivo model; however, these studies were not performed in ovarian cancer xenografts [87]. Therefore, further pre-clinical research is needed to determine if this combinatorial treatment is efficacious in BRCA-proficient EOC patients. 

#### 3.3.2. Clinical Studies

The phase I/II study TOPACIO/Keynote-162 (NCT02657889) has provided evidence that combinatorial treatment with niraparib and pembrolizumab is promising for platinum-resistant ovarian cancer (PROC). Of 60 evaluable patients [89], 64% had a platinum free interval (PFI) of less than 6 months, 19% were defined as platinum refractory (PFI < 30 days), and 17% were platinum sensitive (PFI > 6 months). Of the total 60 patients, the reported ORR was 25%, with a disease control rate (DCR) of 68%. Of the total 14 observed responses, 11 patients were platinum resistant, two were platinum refractory, and one patient had platinum-sensitive disease. In addition, no new safety concerns were identified in the combinatorial treatment [89]. When specifically examining the 11 patient BRCA mutated cohort, the ORR and DCR were found to be 45% and 73%, respectively [89]. Furthermore, an additional phase II study is currently examining the efficacy of nivolumab and rucaparib in both BRCA wild-type (WT) and mutated EOC patient cohorts (NCT03824704). Interestingly, a recent study by Färkkilä et al. performed immunogenomic profiling from patient enrolled in the TOPACIO/Keynote-162 trial [90]. The group uncovered a mutational signature as a marker of treatment response, including homologous recombination repair defectivity and a positive immune score indicative of interferon-primed exhausted T cells [90]. Patients that possessed one or both of these features exhibited a response to treatment, while those lacking both characteristics demonstrated no response [90]. Moreover, the authors determined through single-cell spatial analysis that two extreme responders had unique clustering of exhausted CD8+ T cells with PD-L1+ macrophages or tumor cells with genomic amplification of *PD-1* and *PD-L1* [90]. This study exemplifies how to use novel techniques to better determine patient selection for clinical treatment regimens. Further sophisticated translational studies for all PD-1 combinatorial treatments are necessary to uncover improved biomarkers to detect EOC patient immune response. 

There have been a multitude of clinical trials initiated in the frontline setting for PARP and PD-1/PD-L1 inhibitors. BGOG/ENGOT Ov43, is a large phase III trial evaluating platinum-based chemotherapy with or without pembrolizumab, followed by maintenance placebo or olaparib in non-BRCA mutated patients (NCT03740165), with PFS and OS as primary study endpoints. ENGOT-Ov44, another phase III study, is currently examining the efficacy of the frontline combination of a PD-L1 inhibitor (TSR-042) and platinum chemotherapy, followed by maintenance niraparib and TSR-042 (NCT03602859) in all patients, regardless of BRCA status. In addition, this study will perform PFS assessments on patients based on tumoral PD-L1 positivity. Finally, the ATHENA phase III trial seeks to evaluate the survival benefit of nivolumab and rucaparib in the maintenance setting, following first-line treatment with platinum-based chemotherapy in patients with any BRCA status (NCT03522246). In addition to these three studies examining the combination of PARP and PD-1/PD-L1 inhibitors with frontline chemotherapy, ENGOT Ov46 seeks to evaluate the combination of bevacizumab, durvalumab, and olaparib with platinum-based chemotherapy (NCT03737643), with bevacizumab being optional according to local practice in the BRCA-mutated arm. Moreover, the JAVELIN PARP 100 study (NCT03642132) also sought to examine the frontline combination of anti-angiogenic, PARP, and PD-1-based therapy; however, in March of 2019 this phase III study was discontinued due to the interim results of the JAVELIN 100 study.

Recurrent studies with additional agents targeting PD-1 and PARP in combination have also exhibited efficacious responses. A phase Ib study evaluated the anti-PD-1 antibody, BGB-A317, and a PARP inhibitor, BGB-290, in solid tumors (NCT02660034) and observed that seven out of 38 patients achieved a PR (five of which were EOC patients), and one EOC patient exhibited a CR [91]. The MEDIOLA trial (NCT02734004) examined the efficacy of combined PD-L1 and PARP inhibition in relapsed platinum sensitive BRCA-mutated EOC patients, and observed a DCR of 81% at 12 weeks and an ORR of 63% [92]. In addition, patients that had 1–2 prior chemotherapeutic lines had an enhanced ORR of 68%, with six of these 22 patients achieving a CR [92]. Lee et al. evaluated a phase II cohort of BRCA-mutated and WT EOC patients with recurrent disease treated with a combination of durvalumab and olaparib (NCT2484404) and discovered a DCR of 53% [93]. These results demonstrated clinical activity of durvalumab and olaparib, particularly in BRCA WT patients that had multiple lines of chemotherapy, with biomarker evaluation studies ongoing [93]. Furthermore, this trial along with two additional phase II investigation (NCT02873962, NCT03574779) are currently in progress evaluating treatment arms that contain anti-angiogenic, PARP, and PD-1/PD-L1-based therapy. A comprehensive list of clinical trials evaluating anti-PD-1 and PARP combinatorial therapy can be seen in Table 3. 

Unsurprisingly, pre-clinical and clinical studies have demonstrated extraordinary promise for combinatorial regimens containing PARP and PD-1/PD-L1 inhibitors in EOC patients harboring BRCA mutations. Efficacy for non-BRCA patients is largely uncertain with many trials uncompleted; however, it is likely that the addition of chemotherapy or angiogenic inhibition will be needed to improve response rates. Finally, novel immunogenomic profiling studies have begun to be performed to best determine markers of patient response to PARP and PD-1/PD-L1 dual treatment, which strongly need to be recapitulated in other PD-1-based combinatorial treatment studies.

### 3.4. PD-1 and Additional Immune Receptor Targeting 

#### 3.4.1. Preclinical Studies 

In EOC, it has been proposed that in order to achieve maximal benefit from anti-PD-1 therapy, targeting an additional immune receptor is necessary [94]. Therefore, there have been numerous preclinical studies that have examined the efficacy of targeting PD-1 in combination with either immune receptor inhibitors or agonists. 

There have been three studies that have examined PD-1 blockade in combination with immune receptor agonists. The first study by Wei et al. examined the efficacy of an anti-PD-1 antibody with a CD137 agonist in vivo and found that combinatorial treatment doubled survival of the mice [95]. In addition, mice treated with the dual therapy exhibited significantly higher levels of CD8+ T cells in the peritoneal cavity and spleen, which were determined to have enhanced functionality due to increased cytolytic activity and IFNγ production. The addition of cisplatin to this dual therapy further improved survival by over 90 days [95]. Guo et al. determined that a combination of PD-1 blockade with an OX40 agonist effectively inhibited tumor growth, with 60% of mice declared tumor free post-treatment, while neither monotherapy achieved anti-tumor efficacy [96]. Moreover, dual targeting increased CD4+ and CD8+ T cells and decreased T_reg_ and MDSCs in the peritoneal cavity [96]. Finally, a study by Lu et al. tested dual anti-PD-1 therapy in combination with targeting glucocorticoid induced tumor necrosis factor receptor related protein (GITR), which acts to increase T cell proliferation, activation, and cytokine production [97]. In a syngeneic ID8 murine model, the authors found a synergistic reduction in tumor growth from this combinatorial therapy. Additionally, the dual treatment produced increased levels of IFNy producing effector T cells, while at the same time decreasing immunosuppressive factors such as T_regs_ and MDSC, promoting an immune responsive TME [97]. Furthermore, the authors observed that adding either paclitaxel or cisplatin to this regimen further improved antitumor efficacy [97]. 

Studies examining PD-1’s potential to synergize with an additional immune checkpoint inhibitor have also been investigated. Sawada et al. found evidence for the potential of dual targeting of PD-1 with immune checkpoint TIM-3, as the cytotoxicity of PD-1+TIM-3+CD8+TILs was significantly lower than that of either PD-1-TIM-3- or PD-1+TIM3-CD8+TILs [98]. Interestingly, an immune profiling study by Rådestad et al. examined expression of multiple immune receptors in EOC and demonstrated that PD-1 and TIM-3 were the most commonly co-expressed immune checkpoint receptors on intratumoral CD8+ T cells, further suggesting the promise of combinatorial targeting of these two factors [99]. While these results are encouraging, EOC in vivo targeting studies of these two immune checkpoints have yet to be performed to demonstrate anti-tumor efficacy.

An in vivo study by Liu et al. uncovered that PD-1 or CTLA-4 monotherapy or combinatorial therapy reduced suppressive potential of PD-1+CTLA-4+ MDSCs, and reduced tumor growth and improved survival [100]. Duraiswamy et al. also examined a dual PD-1 and CTLA-4 blockade and found that the regimen reversed CD8+ T cell dysfunction and produced tumor rejection in two-thirds of mice in an ID8-VEGF mouse model [101]. Moreover, dual blockade led to an overall increase in the proliferation of CD8+ and CD4+ T cells and antigen-specific cytokine release along with the inhibition of immunosuppressive T_reg_ functions. Finally, when the dual treatment was tested in combination with a GVAX vaccine (granulocyte macrophage colony-stimulating factor expressing irradiated tumor cells), it produced tumor rejection in 75% of mice [101]. This same group also tested this vaccine with PD-1 blockade combined with either CD137 or toll like receptor-9 (TLR-9) agonists and observed 75% tumor rejection in the ID8-VEGF model. Furthermore, it was determined that treatment increased proliferation and function of CD8+ T cells and increased effector T cell signaling molecules and memory precursor T cells, while reducing T_regs_ and MDSCs [102]. Lastly, a triple targeting study by Dai et al. discovered that treatment with antibodies targeting PD-1, CTLA-4, and CD137 together extended survival in an ID8 murine EOC model by approximately 50 days, while monotherapies were largely ineffective at prolonging survival [103]. 

In EOC, it has been suggested that targeting the PD-1 axis is of primary importance when considering a dual immune receptor therapy [104]. Huang et al. found that combinatorial PD-1/CTLA-4 blockade or a triple blockade against LAG-3, PD-1, and CTLA-4 resulted in tumor free survival in 20% of mice. Single agent targeting against any single immune checkpoint receptor did not produce a reduction in tumor growth [104]. The authors then performed a dual blockade of LAG-3 and CTLA-4 in PD-1 knockout mice, which resulted in an increase in tumor free survival to 40% of mice, as well as an increase of peritoneal CD8+ T cells and cytokine producing effector T cells, and a decrease in T_regs_ and MDCS [104]. This study emphasizes the authority of the PD-1 pathway in EOC and demonstrates proof-of-principle for the potential development of therapeutics such as CAR-T cells with CRISPR-Cas9 editing of PD-1 [105]. To further support the dominance of the PD-1 pathway in EOC, Imai et al. performed immune profiling in EOC patient ascites via flow cytometry and uncovered that out of all immune receptors studied (PD-1, LAG-3, TIM-3 and BTLA), PD-1 was expressed on the majority of CD4+ and CD8+ T cells, with median levels at 65.8% and 57.7%, respectively [106]. In addition, 72.2% and 68.5% of patients expressed multiple immune checkpoint receptors on CD4+ and CD8+ T cells, respectively, showcasing the high expression of PD-1 on EOC TILs [106]. 

Overall, PD-1-based combinatorial therapies with additional immune receptors have demonstrated extraordinary promise preclinically in EOC, and have propelled several clinical trials. In addition, targeting and profiling studies strongly imply the therapeutic relevance of PD-1 in EOC. 

#### 3.4.2. Clinical Studies 

There have been numerous PD-1-based immune checkpoint combinations studied clinically in EOC, with the majority of investigations centered upon anti-PD-1 and CTLA-4 therapy (NCT02498600, NCT01928394, NCT02923934, NCT01975831, NCT02261220, NCT03026062) (Table 4). Recently, results from the phase II trial NRG GY003 (NCT02498600) evaluating nivolumab alone versus nivolumab and ipilimumab established that patients who received the combinatorial regimen exhibited a higher response rate at 31.4%, compared to 12.2% in the nivolumab alone group [107]. Furthermore, the median PFS in the combination group was 3.9 months, versus 2 months in patients who received nivolumab alone, indicating that while the combination of anti-PD-1 and CTLA-4 therapy enhanced response rate, overall survival benefit is limited and an additional targeted agent may be needed to further improve PFS [107]. While the incidence of grade 3 toxicities was higher with the dual regimen compared to monotherapy (49% and 33%, respectively), overall safety assessments were deemed comparable to previous studies [107]. Finally, it was found that the presence of tumoral PD-L1 did not correlate with response in either treatment groups [107], further indicating the strong need for improved markers of immunotherapy response.

Thus far, there have only been two studies that have explored adding auxiliary agents to anti-PD-1/PD-L1 and CTLA-4-based therapy. A phase I/II trial has been initiated in BRCA mutated patients evaluating if the combination of durvalumab, tremelimumab, and olaparib in the recurrent setting can improve PFS in platinum resistant and sensitive EOC patients (NCT02953457). Additionally, there is phase Ib study examining the efficacy of atezolizumab and ipilimumab in combination with interferon alpha or an anti CD20 antibody (obinutuzumab) (NCT02174172). Results from these studies will be enlightening, as they will aid in determining whether anti-PD-1/PD-L1 and CTLA-4 therapy can be enhanced by additional targeted agents, as well as how this will impact toxicity.

Clinical trials involving combination with PD-1 and other immune receptors have been more limited. A phase II trial is currently ongoing investigating the clinical benefit rate of anti-LAG-3 and PD-1 therapy (NCT03365791), in which results are anticipated given that this dual therapy has demonstrated efficacy preclinically in EOC. Furthermore, a phase I study is currently evaluating use of the oral inhibitor CA170 against PD-L1, PD-L2, and immune checkpoint receptor V-domain Ig suppressor of T cell activation (VISTA) (NCT02812875). Additionally, a study evaluating the safety of pembrolizumab and epacadostat, an indoleamine 2,3 dioxygenase (IDO) inhibitor (NCT03277352), and a B7-H3 monoclonal antibody (NCT02475213) are currently active. Lastly, a phase I/II study evaluating poliovirus receptor related immunoglobin domain containing (PVRIG) alone and in combination with nivolumab has been initiated for solid tumors, including ovarian cancer (NCT03667716). Intriguingly, it has been found that PVRIG’s ligand, poliovirus receptor related-2 (PVRL2) is substantially upregulated in EOC [108,109], indicating the potential efficacy of targeting PVRIG along with PD-1.

As results from only one trial of PD-1 combined with an additional immune receptor have been reported, possible conclusions are limited, but a modest improvement in PFS is indicated. Therefore, this suggests that further agents may need to be added to dual immune checkpoint regimens in order to substantially impact patient survival. Interestingly, despite strong pre-clinical profiling data that has demonstrated rationale for dual targeting TIM-3 and PD-1, these studies have yet to be initiated preclinically or clinically in EOC. Results from future studies evaluating this dual regimen are needed to adequately determine the efficacy of PD-1 with other immune checkpoint targets in EOC.

### 3.5. PD-1 and Novel Immunotherapies 

#### 3.5.1. Preclinical Studies 

Novel immunotherapies including adoptive cell therapy (ACT) and oncolytic viruses (OV) have been employed to enhance PD-1-based therapy in EOC. Gitto et al. observed that in a patient-derived xenograft (PDX) model in which TILs were infused in combination with a PD-1 blockade, mice experienced reduced tumor burden and increased survival [110]. Moreover, Oyer et al. examined the combinatorial treatment of PM21 particle expanded NK cells for ACT with anti-PD-L1 in an NSG (NOD-SCID-IL-2Rγnull) murine EOC model [111]. The rationale for this combination originated from the observation that NK cell ACT resulted in an induction of PD-L1 on ovarian cancer cells. Dual treatment with PD-L1 blockade and NK ACT led to a synergistic reduction in tumor burden, with anti-PD-L1 therapy credited for the maintenance of NK cell cytotoxicity [111]. 

McGray et al. sought to test a tumor antigen armed oncolytic Maraba virus in combination with a PD-1 monoclonal antibody to combat tumor growth, as the group had previously discovered an increase in PD-1+ T cells following monotreatment with the OV [112]. While the authors observed enhanced cytotoxic T cell function, overall levels of TILs and TALs were unchanged. In addition, tumor control was found to be heterogeneous, with some mice exhibiting pseudo-progression, which the authors postulate resulted from an increased inflammatory TME [112]. This varied response reveals the mechanistic complexities of this combinatorial treatment effect on the TME and demands further elucidation. In agreement with this study, Liu et al. found that an oncolytic vaccinia virus produced expression of PD-L1 on both tumor and immune cells [113]. Upon combinatorial treatment with anti-PD-L1 therapy, reduced tumor burden and increased survival in an ID8 mouse model was achieved. Furthermore, they noted CD4+ and CD8+ T cells with increased cytolytic activity characterized by higher expression of IFNγ, granzyme B, and perforin [113]. In addition, dual therapy reduced virus-induced PD-L1+ DCs, MDSCs, TAMs, T_regs_, and exhausted PD-1+CD8+ T cells (113). Another in vivo ID8 mouse study by Kowalsky et al. found that a novel IL-15 superagonist OV combined with a PD-1 monoclonal antibody led to a significant improvement in survival compared to either monotherapy alone [114]. Overall, results from these recent studies suggest that ACT and OV-based therapies will represent an innovative approach to sensitize EOC patients to PD-1-based immunotherapy. However, it will be imperative to first assess the safety and therapeutic efficacy of these combinations in a clinical setting.

#### 3.5.2. Clinical Studies 

In EOC there have been a wealth of trials combining PD-1/PD-L1 inhibition and novel immunotherapeutics (Table 5). Two studies have commenced exploring a regimen of pembrolizumab, aldesleukin (IL-2 therapy), and TILs with cyclophosphamide and/or fluradarabine chemotherapy (NCT01174121, NCT03158935), with an additional study investigating this same combination with ipilimumab (NCT03287674). As these studies involve multiple therapies, evaluating safety and tolerability of these regimens will be key. Moreover, groundbreaking studies investigating the combination of patient personalized engineered immune cells, Vigil therapy, with atezolizumab (NCT03073525) and durvalumab (NCT02725489), have been initiated in gynecological cancers with the primary goal of observing treatment related AEs. Furthermore, a prospective study is examining the hyperthermic treatment, thermotron RF-8, in combination with ACT alone or with the addition of pembrolizumab or chemotherapy (NCT03757858). Results of this study have established acceptable safety profiles and a DCR of 66.7% across all groups; however, no EOC patients were assigned to the arm that included pembrolizumab [115]. 

Cancer vaccines have also begun to be investigated clinically in EOC. Currently, there are two studies evaluating an anti-folate receptor (NCT02764333) and Wilm’s tumor 1 (WT1) vaccine (NCT02737787) in combination with anti-PD-1/PD-L1 therapy in EOC. Safety assessments performed for the combination of anti-PD-L1 therapy and anti-folate receptor vaccine revealed acceptable safety profiles in platinum resistant EOC patients [116]. Furthermore, the dual regimen of a WT1 vaccine and nivolumab was also found to be safe and well tolerated [117]. In addition, the one-year PFS of patients treated with the WT1 vaccine and nivolumab was 64%, which is longer than the one-year PFS of 50% in comparable patient populations. Finally, a high degree of B and T cell specific responses were detected in patients treated with combinatorial therapy, which the study aims to further investigate [117]. Overall, the combination of vaccine and PD-1-based therapy has demonstrated potential in EOC thus far.

A host of studies combining PD-1-based therapies with other targeted immunotherapies are currently ongoing in EOC. A phase I/II study determining the efficacy and safety of PLD combined with durvalumab and motolimod, a toll-like receptor 8 (TLR8) agonist (NCT02431559) demonstrated a similar median PFS between three different arms of dosing withPLD, motolimod, and durvalumab (4.3–5.7 months) and one arm in which patients only received PLD and durvalumab (5.5 months). These results indicate that the addition of a TLR8 agonist is not efficacious in enhancing PLD and anti-PD-L1 therapy. Furthermore, anti-PD-1 therapy has been tested in combination with the anti-CD27 antibody varilumab (NCT02335918). Phase I results determined that the combinatorial treatment was safe and tolerable. Furthermore, three of the 36 patients in all solid tumors achieved a PR (one of which was an EOC patient) and 11 patients exhibited SD [118]. In addition, the study found that out of 27 tumors tested for PD-L1, 24 were deemed negative. Interestingly, the study found that an increase in serum chemokine levels and substantial decrease in circulating T_regs_ represented biomarkers of response to the combinatorial regimen [118]. As phase II of this trial has been initiated for EOC patients, this combination remains promising. Finally, early phase studies with a PD-1 inhibitor with colony stimulating factor 1 receptor (CSFR) inhibitors have been initiated in solid tumors and melanoma (NCT02526017, NCT02452424); however, one of the trials (NCT02452424) has been terminated due to inadequate evidence of clinical efficacy.

PD-1-based therapy combined with novel immunotherapies exhibits potential, especially in regimens containing innovative TIL infusions and cancer vaccines. While early results reported from a small portion of these studies have been encouraging, further data is needed to firmly establish whether these combinations will ultimately be safe and effective for EOC patients. In addition, it is critical that more studies be performed that aim to uncover treatment immune response biomarkers.

### 3.6. PD-1 and Other Agents 

#### 3.6.1. Preclinical Studies

Intriguingly, numerous studies have established that a variety of biological factors have the ability to enhance PD-1/PD-L1 expression. Padmanabhan et al. uncovered that IFNγ induced tumoral PD-L1 and proto-oncogene B-cell lymphoma-3 (BCL3) expression in ovarian cancer cell culture [119]. Additionally, it was hypothesized that PD-L1 expression is influenced by BCL3, as stable BCL3 knockdown cell lines exhibited significantly decreased levels of PD-L1. Furthermore, the authors propose that targeting BCL3 may provide a novel means to modulate PD-L1 expression, as their results suggest that IFNγ induction of PD-L1 is facilitated by BCL3 [119]. In agreement with these results, a study by Zou et al. found that BCL3 promotes IFNγ-induced expression of PD-L1 and that the inhibition of PD-L1 leads to reduced proliferation in BCL3-overexpressing cells, corroborating that PD-L1 is a target of BCL3 [120]. Results from these two studies provide strong rationale to test the combination of a BCL3 agonist and anti-PD-1/PD-L1 therapy in EOC. In addition to BCL3, Guo et al. discovered that dinaciclib, a cyclin-dependent kinase (CDK) inhibitor, induced PD-L1 and PD-L2 levels on TILs, forming the hypothesis that targeting CDKs could potentially improve EOC patient immunotherapeutic response [121]. A recent study by Natoli et al. used a sophisticated in vitro tumor-immune co-culture system (TICSs) consisting of ovarian cancer cells and EOC patient ascites and observed that ovarian cancer chemoresistant cells had overall low levels of human leukocyte antigen (HLA) contributing to a poor response to nivolumab monotherapy [122]. To enhance nivolumab response, the authors found that HLA could be induced by a DNA methyltransferase inhibitor, providing a potential method to enhance patient response to PD-1 blockade [122]. Finally, it was also found that inhibition of histone deacetylase (HDAC) via romidepsin along with IFNγ induces PD-L1 expression [123]. Overall, these investigations provide novel approaches to induce PD-L1 expression in EOC cells.

There have been a host of in vivo EOC preclinical studies that have examined a variety of targeted agents in combination with PD-1. Li et al. used an in vitro three-dimensional microfluidic model to show that treatment with a bromodomain containing 4 (BRD4) specific inhibitor, AZD5153, decreased PD-L1 levels on M2 macrophages [124]. BRD4 is a member of the bromodomain and extra terminal (BET) family which is responsible for the promotion of c-myc, resulting in increased cell proliferation. When AZD5153 was tested in combination with anti-PD-L1 therapy in vivo, dual treatment produced a synergistic reduction in tumor burden [124]. Crawford et al. reported efficacy of the combinatorial treatment of a PD-1 monoclonal antibody with REGN4018, a bispecific antibody against mucin 16 (MUC16), a glycoprotein frequently overexpressed in EOC. Dual treatment inhibited tumor growth to a greater degree than single agent treatment, providing a promising option for patients with high MUC16 expression [125]. 

Remarkably, a study by Wang et al. determined that the glycolytic enzyme, pyruvate dehydrogenase kinase-1 (PDK1) correlates to levels of PD-L1 in EOC patient tissue [126]. It was also discovered that PDK1 overexpression resulted in reduced IFNγ secretion leading to reduced CD8+ T cell function, with a PDK1 knockdown reversing this effect. Furthermore, when PDK1 and PD-L1 were targeted in combination in vivo, an overall increase in survival and IFNγ levels resulted, highlighting the prominent role PDK1 plays in T cell dysfunction through the upregulation of PD-L1, and representing the first mechanistic study in EOC to identify a relationship between the PD-1/PD-L1 pathway and cellular metabolism [126]. Fascinatingly, it was even found that the anti-oxidant resveratrol in combination with a PD-1 monoclonal antibody significantly decreased tumor growth in an ID8 murine model [127]. By and large, outcomes from these preclinical in vivo studies provide a convincing rationale for the clinical testing of novel combinatorial approaches. 

In addition to dual blockades, there have also been two studies that have examined ways to enhance delivery of PD-1-based therapy. Cao et al. examined the efficacy of employing nanomedicine to enhance delivery of anti-PD-1-based therapy. The group used photothermolysis, light absorbing copper sulfide nanoparticles, combined with a toll like receptor-9 agonist and PD-1 monoclonal antibody in an ID8 vivo murine model [128]. It was discovered that this treatment regimen significantly improved survival compared to either single agent alone [128]. An additional study by Teo et al. used folic acid (FA)-modified polyethylenimine (PEI) polymers to improve the uptake of PD-L1 siRNA, as EOC cells overexpress FA. The addition of these polymers to PD-L1 siRNA resulted in an over two-fold increase in T cell killing compared to scrambled controls [129]. Taken as a whole, these diverse studies indicate that PD-1 has the ability to interact with, and be influenced by, a large number of biological pathways. Further research will need to be performed in order to determine which targeted therapies in combination with PD-1 targeting will be most clinically efficacious.

#### 3.6.2. Clinical Studies 

PD-1 has been studied in combination with molecules that participate in a diverse array of biological pathways (Table 6). The KEYNOTE 191 study assessed the efficacy of small molecule inhibitor acalabrutinib, a bruton tyrosine kinase inhibitor, alone and in combination with pembrolizumab in recurrent EOC patients (NCT02537444). Results from the study demonstrated that patients treated with the dual regimen exhibited an overall response of 9.1% (three out of 33 patients) compared to acalabrutinib alone, with a response rate of 2.9% (one out of 35 patients), indicating that this agent will not be efficacious in EOC. In addition, a phase I study has been initiated in patients with EOC and other solid tumors that harbor MAPK pathway alterations testing the combination of anti-PD-1 therapy with a pan RAF inhibitor (LXH254), with the goal of assessing safety and tolerability (NCT02607813). 

Entinostat, an HDAC inhibitor, has been tested in combination with avelumab in recurrent EOC patients (NCT02915523). Recent results have determined that this combination produced no significant difference in PFS compared to avelumab alone. Furthermore, the toxicity in the dual regimen was significantly higher, demonstrated by a higher rate of reported AEs [130]. Interestingly, an additional study performed in ovarian cancer (NCT2178722) demonstrated an ORR of entinostat and pembrolizumab of only 8.1%. A phase Ib/II study is currently evaluating Mirvetuximab Sorvtansine (IMGN853), a folate receptor alpha (FRα) antibody-drug conjugate (ADC) in combination with bevacizumab, carboplatin, PLD, pembrolizumab or bevacizumab and carboplatin in patients with FRα EOC, primary peritoneal, or fallopian tube cancers (NCT02606305). Initial safety assessments deemed the treatment combinations with IMGN853 to be safe and tolerable [131], while analysis of efficacy is ongoing. A study examining dual therapy of an anti-PD-L1 therapy combined with novel stereotactic ablative radiotherapy (SAbR), was recently terminated due to low accrual (NCT03312114). Finally, a phase I/II study is evaluating the safety of ETC-1922, a Wnt inhibitor, alone and in combination with pembrolizumab in EOC and other solid tumors (NCT02521844).

Intriguingly, there appears to be a disconnect in terms of agents that have been studied in combination with PD-1/PD-L1 inhibitors in pre-clinical versus clinical studies. As many pre-clinical results have mechanistically identified molecules, such as BCL3, HLA, and CDKs that when targeted enhance PD-1 or PD-L1 expression, it is logical to test these dual regimens clinically, rather than focus on combinatorial therapies that lack strong pre-clinical validations.

## 4. Discussion

Among the broad range of chemotherapeutic and biologic agents that have been tested in combination with PD-1-based therapy in EOC, several common themes have begun to emerge. At the preclinical level, there is a critical deficiency in the number of investigations that elucidate synergistic mechanisms that exist between potentially efficacious PD-1-based combinatorial therapies in EOC. Emphasis on these specific preclinical studies will allow for an improved understand of the complex mechanistic interplay that exists between PD-1/PD-L1 inhibitors and other therapeutic agents, and the ultimate impact this exerts on the TME and diverse cancer signaling pathways. As a result, data generated from these investigations can be exploited to modify or enhance currently tested PD-1-based combinatorial regimens to improve patient response. Moreover, in order to effectively study synergistic mechanisms between PD-1/PD-L1 inhibitors and targets, superior preclinical in vivo models more indicative of the ovarian TME need to be employed. By and large, the majority of pre-clinical studies reviewed used ID8 in vivo models to examine efficacy of PD-1 combinatorial regimens. This is problematic as the ID8 model does not adequately recapitulate human EOC, as it does not contain a *p53* mutation that is inherent in 94% of human EOC [132]. While ID8 *p53*, *VEGF*A, and *Def29* mutated cell lines are examples of more accurate representations of human EOC, only a handful of studies in this review employed these cell lines. Furthermore, the ID8 model has been deemed far less immunogenomic than the spontaneously transformed ovarian surface epithelial (STOSE) model [133]. McClosky et al. generated a novel STOSE model and observed similar growth rates, genomic profile, and immunohistochemical markers (pan-CK+, WT1+, inhibin-, and Pax+) to human high-grade serous ovarian cancer [134]. In addition, STOSE tumors demonstrate increased T cell activation and T_reg_ levels compared to ID8 tumors [133], providing evidence that newer models would be more appropriate to test PD-1-based combinatorial therapies. Finally, non SCID gamma (NSG) mouse lines provide a superior method that permits the engraftment of human tumors. However, the major drawback of this model is its high cost [135], making widespread use limited. 

While many large EOC clinical trials are ongoing, results that have been reported strongly indicate that additional agents will need to be added to a PD-1-based dual regimen to increase EOC patient survival. Anticipated studies evaluating the safety and tolerability of such regimens will be key in determining the feasibility of including multiple agents in frontline or recurrent settings. The principal barrier facing PD-1-based combinatorial treatments clinically is the inability to effectively select patients that will be most responsive to therapy, as results from multiple studies have confirmed the inadequacy of using tumoral PD-L1 expression as a marker of therapeutic response. Overall, a substantial increase in the amount of translational studies are needed as companions to clinical trials to uncover both circulating and tumoral predictive markers of response. The emergence of novel profiling technologies such as multiplexed immunohistochemistry, high parameter flow cytometry, and single cell sequencing represent powerful tools that can be employed to uncover unique predictive signatures to individualize PD-1-based combinatorial treatments [136]. A summarized depiction of both preclinical and clinical areas of focus to improve EOC patient response to PD-1-based combinatorial regimens can be seen in Figure 1.

## 5. Conclusions

Taken as a whole, while the outlook for PD-1-based combinatorial therapeutics in EOC is promising, it is not without challenges. Future mechanistic studies to uncover how various combinatorial treatments affect the TME and translational approaches to identify predictive biomarkers will ultimately be integral in developing successful PD-1-based combinatorial therapies for EOC patients. 

## Figures and Tables

**Figure 1 cancers-12-02150-f001:**
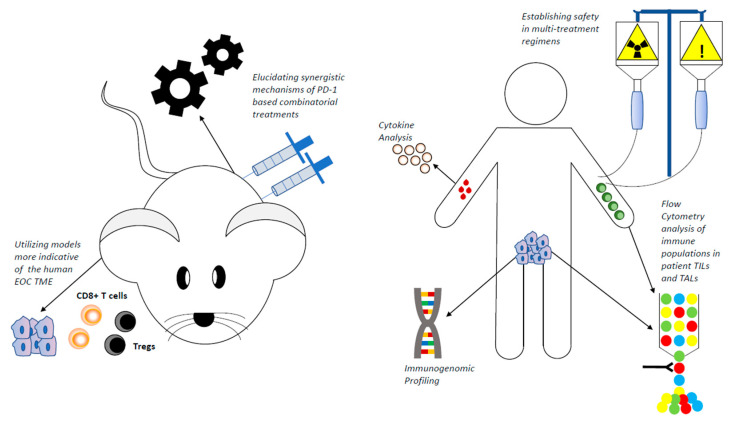
Preclinical and clinical areas of focus to enhance PD-1-based combinatorial treatment response.

**Table 1 cancers-12-02150-t001:** PD-1 and chemotherapy.

Study Title	Trial Identifier	Study Arm(s)	Primary Endpoint(s)	Results
A Phase 1b to Assess the Safety and Tolerability of Carboplatin-Cyclophosphamide Combined with Atezolizumab, an Antibody That Targets Programmed Death Ligand 1 (PD-L1), in Patients with Advanced Breast Cancer and Gynecologic Cancer	NCT02914470	Single Arm	Toxicity; AEs	N/A
Matched Paired Pharmacodynamics and Feasibility Study of Durvalumab in Combination with Chemotherapy in Frontline Ovarian Cancer (N-Dur)	NCT02726997	Single Arm	Pharmacodynamic changes	N/A
Anti-PD-1 Therapy in Combination with Platinum Chemotherapy for Platinum Resistant Ovarian, Fallopian Tube, and Primary Peritoneal Cancer	NCT03029598	Single Arm	PFS	N/A
A Phase II Study of Pembrolizumab With Cisplatin and Gemcitabine Treatment in Patients with Recurrent Platinum-Resistant Ovarian Cancer	NCT02608684	Single Arm	ORR by RECIST	CR: 5.6% PR: 55.6% SD: 27.8% PD: 11.1%
Phase II Open Label Nonrandomized Trial of the Anti PD-1 Therapy Pembrolizumab with First Line Platinum-Based Chemotherapy Followed by 12 Months Pembrolizumab Monotherapy for Patients with Stage III/IV Epithelial Ovarian Cancer	NCT02766582	Single Arm	PFS	N/A
Matched Paired Pharmacodynamics and Feasibility Study of Pembrolizumab in Combination with Chemotherapy in Frontline Ovarian Cancer	NCT02520154	Single Arm	PFS	N/A
Phase II Single Arm Study of Combination Pembrolizumab, Paclitaxel, and Carboplatin in Patients with Advanced Stage Ovarian, Fallopian Tube, or Peritoneal Carcinoma Receiving Neoadjuvant Chemotherapy	NCT02834975	Single Arm	pORR	N/A
A Phase II Study of Pembrolizumab Combined with Pegylated Liposomal Doxorubicin (PLD) for Recurrent Platinum Resistant Ovarian, Fallopian Tube or Peritoneal Cancer	NCT02865811	Single Arm	CBR	ORR: 11.5% [72]
A Randomized, Open-label, Multicenter, Phase 3 Study to Evaluate the Efficacy and Safety of Avelumab (MSB0010718C) in Combination with and/or Following Chemotherapy in Patients with Previously Untreated Epithelial Ovarian Cancer JAVELIN OVARIAN 100	NCT02718417	Arm 1: Chemotherapy Arm 2: Chemotherapy + avelumab maintenance Arm 3: Chemotherapy + avelumab + avelumab maintenance	PFS	Arm 1: N/A Arm 2: PFS: 16.8 months Arm 3: PFS: 18.1 months Study Terminated
A Phase 3, Multicenter, Randomized, Open-label Study of Avelumab (MSB0010718C) Alone or in Combination with Pegylated Liposomal Doxorubicin Alone in Patients with Platinum-Resistant/Refractory Ovarian Cancer	NCT02580058	Arm 1: Avelumab Arm 2: Avelumab + PLD Arm 3: PLD	OS; PFS	Arm 1: OS: 11.8 months PFS: 1.9 months Arm 2: OS: 15.7 months PFS: 3.7 months Arm 3: OS: 13.1 months PFS: 3.5 months

AEs, adverse events; CBR, clinical benefit rate; CR, complete response; ORR, objective response rate; OS, overall survival; PD, progressive disease; PFS, progression-free survival; PLD, pegylated liposomal doxorubicin; pORR, pathologic objective response rate; PR, partial response; RECIST, response evaluation criteria in solid tumors; SD, stable disease.

**Table 2 cancers-12-02150-t002:** PD-1 and anti-angiogenic therapy.

Study Title	Trial Identifier	Study Arm(s)	Primary Endpoint(s)	Results
Phase Ib Trial of Pembrolizumab and Nintedanib	NCT02856425	Single Arm	MTD of Nintedanib	N/A
A Phase II Study of the Anti-PDL1 Antibody Atezolizumab, Bevacizumab and Acetylsalicylic Acid to Investigate Safety and Efficacy of This Combination in Recurrent Platinum-Resistant Ovarian, Fallopian Tube or Primary Peritoneal Adenocarcinoma	NCT02659384	Arm 1: Bevacizumab Arm 2: Atezolizumab + Bevacizumab Arm 3: Atezolizumab + Bevacizumab + Acetylsalicylic Acid	PFS	N/A
A Phase II Study with a Safety Lead-in of Nivolumab in Combination with Bevacizumab or in Combination with Bevacizumab and Rucaparib for the Treatment of Relapsed Epithelial Ovarian, Fallopian Tube or Peritoneal Cancer	NCT02873962	Arm 1: Nivolumab + Bevacizumab Arm 2: Nivolumab + Bevacizumab + Rucaparib	ORR by RECIST	ORR: 28.9% (40% platinum-sensitive 16.7% platinum- resistant) [79]
A Randomized, Phase II/III Study of Pegylated Liposomal Doxorubicin and CTEP-Supplied Atezolizumab Versus Pegylated Liposomal Doxorubicin/Bevacizumab and CTEP-Supplied Atezolizumab Versus Pegylated Liposomal Doxorubicin/Bevacizumab in Platinum-Resistant Ovarian Cancer	NCT02839707	Arm 1: PLD + Atezolizumab Arm 2: PLD + Bevacizumab + Atezolizumab Arm 3: PLD + Bevacizumab	DLT; PFS; OS	N/A
Atezolizumab in Combination with Bevacizumab and Chemotherapy versus Bevacizumab and Chemotherapy in Recurrent Ovarian Cancer—ARandomized Phase III Trial	NCT03353831	Arm 1: Chemotherapy + Bevacizumab Arm 2: Chemotherapy + Bevacizumab + Atezolizumab	OS; PFS	N/A
A Phase III, Multicenter, Randomized, Study of Atezolizumab versus Placebo Administered in Combination with Paclitaxel, Carboplatin, and Bevacizumab to Patients with Newly Diagnosed Stage III or Stage IV Ovarian, Fallopian Tube, or Primary Peritoneal Cancer	NCT03038100	Arm 1: Atezolizumab + Paclitaxel, Carboplatin and Bevacizumab Arm 2: Paclitaxel, Carboplatin and Bevacizumab	PFS, OS	N/A
A Randomized, Double-Blinded, Phase III Study of Atezolizumab versus Placebo in Patients with Late Relapse of Epithelial Ovarian, Fallopian Tube, or Peritoneal Cancer Treated by Platinum-Based Chemotherapy and Bevacizumab	NCT02891824	Arm 1: Bevacizumab + Chemotherapy Arm 2: Atezolizumab + Bevacizumab + Chemotherapy	PFS	N/A

DLT, dose-limiting toxicities; MTD, maximum tolerated dose; ORR, objective response rate; OS, overall survival; PFS, progression-free survival; PLD, pegylated liposomal doxorubicin; RECIST, response evaluation criteria in solid tumors.

**Table 3 cancers-12-02150-t003:** PD-1 and anti-PARP Therapy.

Study Title	Trial Identifier	Study Arm(s)	Primary Endpoint(s)	Results
A Phase 1/1b, Open Label, Multiple Dose, Dose Escalation and Expansion Study to Investigate the Safety, Pharmacokinetics and Antitumor Activity of the Anti-PD-1 Monoclonal Antibody BGB-A317 in Combination with the PARP Inhibitor BGB-290 in Subjects with Advanced Solid Tumors	NCT02660034	Single Arm	Phase 1: AEs; DLT; MTD Phase 1b: ORR, PFS DOR, DCR, CBR, OS	PR: 7(5 EOC)/38 patients CR:1(EOC)/38 patients [89]
A Phase I/II Evaluation of Olaparib in Combination with Durvalumab (Medi4736) and Tremelimumab in the Treatment of Recurrent Platinum-Sensitive or Resistant or Refractory Epithelial Ovarian, Fallopian Tube, or Primary Peritoneal Cancer in Patients Who Carry a BRCA1 or BRCA2 Mutation	NCT02953457	Single Arm	DLT, PFS	N/A
Phase I/II Study of the Anti-Programmed Death Ligand-1 Antibody MEDI4736 in Combination with Olaparib and/or Cediranib for Advanced Solid Tumors and Advanced or Recurrent Ovarian, Triple Negative Breast, Lung, Prostate and Colorectal Cancers	NCT02484404	Sequential Assignment	Phase I: RP2D Phase II: ORR	DCR: 53% [91]
Phase 1/2 Clinical Study of Niraparib in Combination with Pembrolizumab (MK-3475) in Patients with Advanced or Metastatic Triple-Negative Breast Cancer and in Patients with Recurrent Ovarian Cancer	NCT02657889	Single Arm	Phase I: DLT, AEs Phase II: ORR	ORR:25%, [87]
A Phase I/II Study of MEDI4736 (Anti-PD-L1 Antibody) in Combination with Olaparib (PARP Inhibitor) in Patients with Advanced Solid Tumors	NCT02734004	Single Arm	DCR, Safety and tolerability, ORR	DCR:81% ORR:63% [90]
A Phase 2, Open-label Study to Evaluate Rucaparib in Combination with Nivolumab in Patients with Selected Solid Tumors (ARIES)	NCT03824704	Single Arm	ORR	N/A
Phase 2 Multicohort Study to Evaluate the Safety and Efficacy of Novel Treatment Combinations in Patients with Recurrent Ovarian Cancer	NCT03574779	Single Arm	ORR	N/A
ATHENA (A Multicenter, Randomized, Double-Blind, Placebo-Controlled Phase 3 Study in Ovarian Cancer Patients Evaluating Rucaparib and Nivolumab as Maintenance Treatment Following Response to Front-Line Platinum-Based Chemotherapy)	NCT03522246	Arm 1: Rucaparib + nivolumab Arm 2: Rucaparib Arm 3: Nivolumab Arm 4: Placebo	PFS	N/A
ENGOT-0V44 The FIRST (First-Line Ovarian Cancer Treatment with Niraparib Plus TSR-042) Study: A Randomized, Double-Blind, Phase 3 Comparison of Platinum-Based Therapy with TSR-042 and Niraparib Versus Standard of Care Platinum-Based Therapy as First-Line Treatment of Stage III or IV Nonmucinous Epithelial Ovarian Cancer	NCT03602859	Arm 1: Chemotherapy + Bevacizumab Arm 2: Chemotherapy + Niraparib Arm 3: Chemotherapy + TSR-042	PFS	N/A
A Randomized Phase 3, Double-Blind Study of Chemotherapy with or without Pembrolizumab Followed by Maintenance with Olaparib or Placebo for the First-Line Treatment of BRCA Non-Mutated Advanced Epithelial Ovarian Cancer (EOC) (KEYLYNK-001/ENGOT-ov43)	NCT03740165	Arm 1: Pembrolizumab + Olaparib Arm 2: Pembrolizumab Arm 3: Placebo	PFS; OS	N/A
A Randomized, Open-Label Multicenter, Phase 3 Study to Evaluate the Efficacy and Safety of Avelumab in Combination with Chemotherapy Followed by Maintenance Therapy of Avelumab in Combination with the Poly (Adenosine Diphosphate [ADP]-Ribose) POLYMERASE (PARP) Inhibitor Talazoparib in Patients with Previously Untreated Advanced Ovarian Cancer (JAVELIN OVARIAN PARP100)	NCT03642132	Arm 1: Chemotherapy + Avelumab + Talazoparib Arm 2: Chemotherapy + Talazoparib Arm 3: Chemotherapy + Bevacizumab	PFS	Study Discontinued
A Phase III Randomized, Double-Blind, Placebo-Controlled, Multicenter Study of Durvalumab in Combination with Chemotherapy and Bevacizumab, Followed by Maintenance Durvalumab, Bevacizumab and Olaparib in Newly Diagnosed Advanced Ovarian Cancer Patients (DUO-O).	NCT03737643	Arm 1: Chemotherapy + Bevacizumab + maintenance Bevacizumab, Arm 2: Chemotherapy + Bevacizumab + Durvalumab + maintenance Bevacizumab + Durvalumab Arm 3: Chemotherapy + Bevacizumab +Durvalumab + maintenance Bevacizumab + Durvalumab + Olaparib. tBRCAm cohort: Chemotherapy + Bevacizumab + Durvalumab + maintenance Bevacizumab + Durvalumab + Olaparib.	PFS	N/A

AEs, adverse events; CBR, clinical benefit rate; CR, complete response; DCR, disease control rate; DLT, dose-limiting toxicities; DOR, duration of response; EOC, epithelial ovarian cancer; MTD, maximum tolerated dose; ORR, objective response rate; OS, overall survival; PFS, progression-free survival; PR, partial response; RP2D, recommended phase 2 dose.

**Table 4 cancers-12-02150-t004:** PD-1 and additional immune receptors.

Study Title	Trial Identifier	Study Arm(s)	Primary Endpoint(s)	Results
A Phase 1 Study to Evaluate the Safety and Tolerability of Anti-PD-L1, MEDI4736, in Combination with Tremelimumab in Subjects with Advanced Solid Tumors	NCT01975831	Single Arm	AEs	N/A
A Phase 1, Open-Label, Dose Escalation and Dose Expansion Trial Evaluating the Safety, Pharmacokinetics, Pharmacodynamics, and Clinical Effects of Orally Administered CA-170 in Patients with Advanced Tumors and Lymphomas	NCT02812875	Single Arm	DLT; MTD; RP2D	N/A
A Phase I Study of MEDI4736 (Anti-PD-L1 Antibody) in Combination with Tremelimumab (Anti-CTLA-4 Antibody) in Subjects with Advanced Solid Tumors	NCT02261220	Single Arm	AEs, SAEs, DLT, ORR	N/A
A Phase 1, Open-Label, Dose Escalation Study of MGA271 in Combination with Pembrolizumab and in Combination with MGA012 in Patients with Melanoma, Squamous Cell Cancer of the Head and Neck, Non-Small Cell Lung Cancer, Urothelial Cancer, and Other Cancers	NCT02475213	Sequential Assignment	AEs; SAEs	N/A
A Phase 1a/1b Study of COM701 as Monotherapy and in Combination with an Anti-PD-1 Antibody in Subjects with Advanced Solid Tumors	NCT03667716	Arm 1: Monotherapy dose escalation Arm 2: Combination dose escalation Arm 3: Monotherapy expansion Arm 4: Combination dose expansion	AEs; MTD	N/A
A Phase Ib Study of the Safety and Pharmacology of Atezolizumab (Anti−Pd-L1 Antibody) Administered with Ipilimumab, Interferon-Alpha, or Other Immune-Modulating Therapies in Patients with Locally Advanced or Metastatic Solid Tumors	NCT02174172	Arm 1: Atezolizumab + Ipilimumab Arm 2: Atezolizumab + Interferon alfa-2 Arm 3: Atezolizumab + Peg- Interferon alfa-2a Arm 4: Atezolizumab + Peg-Interferon alfa-2a+ Bevacizumab Arm 5: Atezolizumab + Obinutuzumab	RP2D; AEs	N/A
A Phase 1/2, Open-Label Study of Nivolumab Monotherapy or Nivolumab Combined with Ipilimumab in Subjects with Advanced or Metastatic Solid Tumors	NCT01928394	Arm 1: Nivolumab Arm 2–5: Nivolumab + Ipilimumab at various dosages Arm 6: Nivolumab + Ipilimumab + Cobimetinib	ORR	OC Arm N-I Dose Level 2: ORR: 12.2% AEs: 58.54% OC Arm N-I Dose Level 2b: ORR: 7.0% AEs: 72.09% OC Arm N-I Dose Level 2c: ORR: 9.5% AEs: 69.05%
Phase II Randomized Trial of Nivolumab with or without Ipilimumab in Patients with Persistent or Recurrent Epithelial Ovarian, Primary Peritoneal, or Fallopian Tube Cancer	NCT02498600	Arm 1: Nivolumab Arm 2: Nivolumab + Ipilimumab	ORR	ORR: 12.2% (Arm 1), 31.4% (Arm 2) PFS: 2 mo. (Arm 1), 3.9 mo. (Arm 2) [105]
A Phase II Clinical Trial Evaluating Ipilimumab and Nivolumab in Combination for the Treatment of Rare Gastrointestinal, Neuro-Endocrine and Gynecological Cancers	NCT02923934	Single Arm	CBR	N/A
Modular Phase 2 Study to Link Combination Immune-Therapy to Patients with Advanced Solid and Hematologic Malignancies. Module 9: PDR001 Plus LAG525 for Patients with Advanced Solid and Hematologic Malignancies.	NCT03365791	Single Arm	CBR; PFS	N/A
Randomized Phase II Trial of Durvalumab (MEDI4736) and Tremelimumab Administered in Combination versus Sequentially in Recurrent Platinum-Resistant Epithelial Ovarian Cancer	NCT03026062	Arm 1: Sequential Tremelimumab + Durvalumab Arm 2: Combination Tremelimumab + Durvalumab	irPFS	N/A

AEs, adverse events; CBR, clinical benefit rate; CRR, complete response rate; DLT, dose-limiting toxicities; irPFS, immune-related progression-free survival; MTD, maximum tolerated dose; OC, ovarian cancer; ORR, objective response rate; PFS, progression-free survival; RP2D, recommended phase 2 dose; SAEs, serious adverse events.

**Table 5 cancers-12-02150-t005:** PD-1 and novel immunotherapies.

Study Title	Trial Identifier	Study Arm(s)	Primary Endpoint(s)	Results
A Phase I Study of Concomitant WT1 Analog Peptide Vaccine with Montanide and GM-CSF in Combination with Nivolumab in Patients with Recurrent Ovarian Cancer Who Are in Second or Greater Remission	NCT02737787	Single Arm	DLT	1-year PFS: 64.0% [115]
A Phase 1a/1b Study of Cabiralizumab in Combination with Nivolumab in Patients with Selected Advanced Cancers	NCT02526017	Arm 1: Cabiralizumab Arm 2: Cabiralizumab + Nivolumab	AEs; SAEs; RD; ORR	N/A
Phase Ib Trial of Pembrolizumab Administered in Combination with or Following Adoptive Cell Therapy—A Multiple Cohort Study; the ACTIVATE (Adoptive Cell Therapy InVigorated to Augment Tumor Eradication) Trial	NCT03158935	Arm 1: Cyclophosphamide and fludarabine followed by TILs, IL-2, and Pembrolizumab (Advanced metastatic melanoma) Arm 2: Cyclophosphamide followed by TILs, IL-2, and Pembrolizumab (Advanced ovarian cancer)	SAEs	N/A
A Prospective Study of Hyperthermia Combined with Autologous Adoptive Cellular Immunotherapy in the Treatment of Abdominal and Pelvic Malignancies or Metastases	NCT03757858	Arm 1: HT + ACT Arm 2: HT + ACT + Pembrolizumab Arm 3: HT + ACT + CT Arm 4: HT + CT	AEs; ORR	ORR: 30.0% DCR: 66.7% [113]
T-Cell Therapy in Combination with Checkpoint Inhibitors for Patients with Advanced Ovarian, Fallopian Tube and Primary Peritoneal Cancer	NCT03287674	Single Arm	AEs	N/A
A Phase I/II Dose Escalation and Cohort Expansion Study of the Safety, Tolerability and Efficacy of Anti-CD27 Antibody (Varlilumab) Administered in Combination with Anti-PD-1 (Nivolumab) in Advanced Refractory Solid Tumors	NCT02335918	Single Arm	Phase I: AEs Phase II: ORR; OS	PR:3(1 EOC)/36 patients SD: 11/36 patients [116]
Phase 1/2 Study of Chemoimmunotherapy with Toll-like Receptor 8 Agonist Motolimod (VTX-2337) + Anti-PD-L1 Antibody MEDI4736 in Subjects with Recurrent, Platinum-Resistant Ovarian Cancer for Whom Pegylated Liposomal Doxorubicin is Indicated	NCT02431559	Arm 1: Durvalumab + PLD + motolimod dose 0a Arm 2: Durvalumab + PLD + motolimod dose 0b Arm 3: Durvalumab + PLD + motolimod dose level 1 Arm 4: Durvalumab + PLD	AEs; PFS	Arm 1: SAEs: 0.0% AEs: 100.0% PFS at 6 months: 33.3% PFS: 5.6 mo. CR: 0.0% PR: 33.3% SD: 33.3% PD: 33.3% Arm 2: SAEs: 50.0% AEs: 100.0% PFS at 6 months: 50.0% PFS: 5.7 mo. CR: 0.0% PR: 25.0% SD: 25.0% PD: 50.0% Arm 3: SAEs: 16.7% AEs: 100.0% PFS at 6 mo.: 33.3% PFS: 4.3 mo. CR: 0.0% PR: 33.3% SD: 16.7% PD: 50.0% Arm 4: SAEs: 57.5% AEs: 100.0% PFS at 6 mo.: 42.9% PFS: 5.5 mo. CR: 5.0% PR: 10.0% SD: 45.0% PD: 40.0%
Phase 1/2a Study of Double-Immune Suppression Blockade by Combining a CSF1R Inhibitor (PLX3397) with an Anti-PD-1 Antibody (Pembrolizumab) to Treat Advanced Melanoma and Other Solid Tumors	NCT02452424	Single Arm	AEs	Study Terminated
A Phase II Trial of TPIV200/huFR-1 (A Multi-Epitope Anti-Folate Receptor Vaccine) Plus Anti-PD-L1 MEDI4736 (Durvalumab) in Patients with Platinum Resistant Ovarian Cancer	NCT02764333	Single Arm	ORR	Acceptable Combinatorial Safety Profile (114)
A Randomized, Intra-Patient Crossover, Safety, Biomarker and Anti-Tumor Activity Assessment of the Combination of Atezolizumab and Vigil in Patients with Advanced Gynecological Cancers	NCT03073525	Arm 1: Vigil + Atezolizumab Arm 2: Vigil and Vigil + Atezolizumab Arm 3: Atezolizumab and Vigil + Atezolizumab Arm 4: Atezolizumab	AEs	N/A
Pilot Study of Durvalumab (MEDI4736) in Combination with Vigil in Advanced Women’s Cancers	NCT02725489	Arm 1: 1 × 10^6^ cells Vigil Arm 2: 1 × 10^7^ cells Vigil Arm 3: 1 × 10^5^ cells Vigil	AEs	N/A
A Phase II Study Using Short-Term Cultured, Autologous Tumor-Infiltrating Lymphocytes Following a Lymphodepleting Regimen in Metastatic Cancers Plus the Administration of Pembrolizumab	NCT01174121	Arm 1: CD8 + TILs + Aldesleukin + Cyclophosphamide + Fludarabine Arm 2: Unselected TILs + Aldesleukin+ Cyclophosphamide + Fludarabine Arm 3: Unselected TILs + Pembrolizumab + Cyclophosphamide Arm 4: Unselected TILs + Cyclophosphamide + Aldesleukin_ Fludarabine + Pembrolizumab	Response rate	N/A

AEs, adverse events; CR, complete response; DCR, disease control rate; DLT, dose-limiting toxicities; EOC, epithelial ovarian cancer; ORR, objective response rate; OS, overall survival; PD, progressive disease; PFS, progression-free survival; PR, partial response; RD, recommended dose; SAEs, serious adverse events; SD, stable disease.

**Table 6 cancers-12-02150-t006:** PD-1 and other agents.

Study Title	Trial Identifier	Study Arm(s)	Primary Endpoint(s)	Results
A Phase I Dose Finding Study of Oral LXH254 in Adult Patients with Advanced Solid Tumors Harboring MAPK Pathway Alterations	NCT02607813	Arm 1: Dose escalation LXH254 Arm 2–4: Dose expansion LXH254 Arm 5: Dose expansion LXH254 + PDR001 Arm 6: Dose escalation LXH254 + PDR001	AEs; DLT	N/A
A Phase 1A/B Study to Evaluate the Safety and Tolerability of ETC-1922159 in Advanced Solid Tumors	NCT02521844	Arm 1: ETC-1922159 + Pembrolizumab Arm 2: ETC-1922159 until progression, then Pembrolizumab added	MTD; RD; AEs	N/A
A Randomized, Placebo-Controlled, Double-Blind, Multicenter Phase 1b/2 Study of Avelumab with or without Entinostat in Patients with Advanced Epithelial Ovarian Cancer which has Progressed or Recurred after First-Line Platinum-Based Chemotherapy and at Least Two Subsequent Lines of Treatment with a Safety Lead-In	NCT02915523	Arm 1: Entinostat + Avelumab Arm 2: Avelumab	AEs; DLT; MTD /RP2D	PFS: 1.64 mo. (Arm 1) 1.51 mo. (Arm 2) AEs (any grade): 93% (Arm 1) 78% Arm (2) [128]
A Phase 1b/2 Study to Evaluate the Safety, Tolerability and Pharmacokinetics of Mirvetuximab Soravtansine (IMGN853) in Combination with Bevacizumab, Carboplatin, Pegylated Liposomal Doxorubicin, Pembrolizumab, or Bevacizumab + Carboplatin in Adults with Folate Receptor Alpha Positive Advanced Epithelial Ovarian Cancer, Primary Peritoneal Cancer or Fallopian Tube Cancer	NCT02606305	Arm 1: Dose escalation and dose expansion IMGN853 +Bevacizumab Arm 2: Dose escalation IMGN853 + Carboplatin Arm 3: Dose escalation IMGN853 + PLD Arm 4: Dose escalation and dose expansion IMGN853 + Pembrolizumab Arm 5: Dose expansion IMGN853 + Bevacizumab + Carboplatin	TEAEs; SAEs; DLT; ORR	IMGN853 Combinations safe and tolerable [129]
A Phase 1/2 Study Exploring the Safety, Tolerability, and Efficacy of MK-3475 in Combination with INCB024360 in Subjects with Selected Cancers (ECHO-202/KEYNOTE-037)	NCT02178722	Single Arm	AEs; ORR	ORR: EOC: 8.1%
Phase II Trial of Concurrent Anti-PD-L1 and SAbR for Patients with Persistent or Recurrent Epithelial Ovarian, Primary Peritoneal or Fallopian Tube Cancer (with Safety Lead-In)	NCT03312114	Single Arm	ORR	Study Terminated
A Phase 2 Proof-of-Concept Study of ACP-196 Alone and in Combination with Pembrolizumab in Subjects with Recurrent Ovarian Cancer	NCT02537444	Arm 1: Acalabrutinib Arm 2: Acalabrutinib + Pembrolizumab	OR	OR: 2.9% (Arm1) 9.1% (Arm 2) SAEs: 21.05% (Arm 1) 41.03% (Arm 2)

AEs, adverse events; DLT, dose-limiting toxicities; EOC, epithelial ovarian cancer; MTD, maximum tolerated dose; OR, overall response; ORR, objective response rate; PFS, progression-free survival; RD, recommended dose; RP2D, recommended phase 2 dose; SAEs, serious adverse events; TEAEs, treatment-emergent adverse events.

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
