# Peer review of "The Perfect Combination: Enhancing Patient Response to PD-1-Based Therapies in Epithelial Ovarian Cancer"

_cancers, 2020, doi:10.3390/cancers12082150_

Round 1

Reviewer 1 Report

In this review article, the authors reviewed comprehensively the up to date, plethora of preclinical studies and clinical trials involving PD1/PD-L1 blockade alone or in combination with other agents in treating ovarian cancer. The preclinical studies and ongoing or completed clinical trials encompassed are fairly complete, thorough, and unbiased. They do provide current knowledge of many therapeutic approaches applied in combination with PD1/PD-L1 blockade regimens--the outcome and perspectives. The readers can appreciate the provided highlights and interpretation of the findings and shortfall of many important studies/trials in this important area of immunotherapies in ovarian cancer. The structure of the manuscript is suitable and the messages are clearly delivered.

I agreed with most of the author’s opinions on barriers to overcome PD-1/PD-L1 blockade therapies, particularly on possible problems and obstacles we face currently regarding the lack of appropriate biomarkers that can be used to better select patients who can be benefited from the PD1/PD-L1 blockade in combination with other agents. Indeed, mechanistic studies on the efficacy of combinatorial therapies utilizing PD-1 blockade with other agents is also critically needed.  

Minor revision:

Line 749-753:  Authors mentioned the disconnect on the outcome of between preclinical and clinical studies.  Perhaps they should elaborate and include a small list of examples mentioned in the manuscript.

Line 717 and 718: the statement " to improve the efficiency of siRNA uptake of PD-L1" was not clear to me.

Author Response

Dear Reviewer #1,

Thank you for taking time to read our manuscript and providing helpful comments and suggestions. Please see below for changes:

  • Line 749-753: Authors mentioned the disconnect on the outcome between preclinical and clinical studies. Perhaps they should elaborate and include a small list of examples mentioned in the manuscript.

While this line refers only to the disconnect in section 3.6 PD-1 and Other Agents we did include a small list of these molecules that we were eluding do which now reads at lines 774-777 “As many pre-clinical results have mechanistically identified molecules, such as BCL3, HLA, and CDKs that when targeted enhance PD-1 or PD-L1 expression, it is logical to test these dual regimens clinically, rather than focus on combinatorial therapies that lack strong pre-clinical validations”.

  • Line 717 and 718: the statement “ to improve the efficiency of siRNA uptake of PD-L1” was not clear to me

The statement now reads : “An additional study by Teo et al utilized folic acid (FA) modified polyethylenimine (PEI) polymers to improve the uptake of PD-L1 siRNA, as EOC cells overexpress FA”, which can be seen on lines 740-741

Reviewer 2 Report

Nicole E James et al in this review summarize and discuss pre-clinical and clinical studies evaluating PD-1 based combinatorial therapies. They discuss the challenges to ovarian cancer therapy caused by immunosuppressive tumor micrenvironment, elevated levels of highly activated T regulatory cells and heterogeneity among patients. Then they deeply show and list all the pre-clinical and clinical studies exploiting the combination of PD-1 with other therapy / immunotherapy to treat ovarian cancer. In this review authors focus their attention on T cells, however it's now clear that NK cells could play an important role in this disease, since the unique ovarian tumor microenvironment, thorough several mechanisms, suppress almost completely their cytotoxic anti tumor effect. This could be really important considering that, as authors say, in ovarian cancer microenvironment there is a strong presence of Treg, while CD8+ cytotoxic T cells are almost missing. On the other hand it is of note that, especially in ascites, NK cells represent an important population among lymphocytes. In addition, it has been shown that NK cells from ascites are characterized by high level of PD-1 (in this context please add appropriate references). Thus PD-1 blockade could exert a positive effect also on NK cells. Thus, they should include some information on NK cells in ovarian cancer, considering that several articles in leterature are hypothesizing an immunotherapy protocol able to restore NK and T cell activity in this gynecologic malignancy.

Minor points:

Please replace INF with IFN to indicate interferon gamma

Author Response

Dear Reviewer #2,

Thank you for taking time to read our manuscript and providing helpful comments and suggestions. Please see below for changes:

  • In this review authors focus their attention on T cells, however it's now clear that NK cells could play an important role in this disease, since the unique ovarian tumor microenvironment, thorough several mechanisms, suppress almost completely their cytotoxic anti tumor effect. This could be really important considering that, as authors say, in ovarian cancer microenvironment there is a strong presence of Treg, while CD8+ cytotoxic T cells are almost missing. On the other hand it is of note that, especially in ascites, NK cells represent an important population among lymphocytes. In addition, it has been shown that NK cells from ascites are characterized by high level of PD-1 (in this context please add appropriate references). Thus PD-1 blockade could exert a positive effect also on NK cells. Thus, they should include some information on NK cells in ovarian cancer, considering that several articles in leterature are hypothesizing an immunotherapy protocol able to restore NK and T cell activity in this gynecologic malignancy.

We greatly appreciate this comment and have added in unique tumor microenvironment section the presence of  functionally impaired NK cells in the ascites that reads : “Malignant ascites produced by the peritoneal cavity are rich in cytokines and growth factors that not only promote tumor growth, but also impair the ability of NK lymphocytes  to function properly as effector cells (23) Furthermore, these impaired NK cells express PD-1 and exhibit a significantly reduced ability to kill PD-1+ tumor cells and secrete IFNγ and TNFα (24)” (Lines 68-71). We completely agree that it is an important point that targeting PD-1 does exert a positive effect on NK cells, unfortunately in the context of the pre-clinical combinatorial PD-1 studies that were reviewed for this paper, effects on NK cells were rarely mentioned. However, when effects of a PD-1 blockade were mentioned in reference to NK cell function, this was reported. For example, please see lines 607-614 from a study by Gitto et al. In addition, we highlight in Figure 1 the importance of future studies that examine response to PD-1 based combinatorial therapies by performing flow cytometry of diverse TAL and TIL populations.

  • Please replace INF with IFN to indicate interferon gamma

INF has been replaced to IFN to indicate interferon gamma in this manuscript.
